# Revisiting the Use of Normal Saline for Peritoneal Washing in Ovarian Cancer

**DOI:** 10.3390/ijms242216449

**Published:** 2023-11-17

**Authors:** Hironari Akasaka, Honami Naora

**Affiliations:** Department of Molecular and Cellular Oncology, University of Texas MD Anderson Cancer Center, Houston, TX 77030, USA

**Keywords:** omentum, peritoneal cavity, neoplasm metastasis, tumor microenvironment, peritoneal lavage, crystalloid solutions

## Abstract

The omentum is the predominant site of ovarian cancer metastasis, but it is difficult to remove the omentum in its entirety. There is a critical need for effective approaches that minimize the risk of colonization of preserved omental tissues by occult cancer cells. Normal saline (0.9% sodium chloride) is commonly used to wash the peritoneal cavity during ovarian cancer surgery. The omentum has a prodigious ability to absorb fluid in the peritoneal cavity, but the impact of normal saline on the omentum is poorly understood. In this review article, we discuss why normal saline is not a biocompatible solution, drawing insights from clinical investigations of normal saline in fluid resuscitation and from the cytopathologic evaluation of peritoneal washings. We integrate these insights with the unique biology of the omentum and omental metastasis, highlighting the importance of considering the absorptive ability of the omentum when administering agents into the peritoneal cavity. Furthermore, we describe insights from preclinical studies regarding the mechanisms by which normal saline might render the omentum conducive for colonization by cancer cells. Importantly, we discuss the possibility that the risk of colonization of preserved omental tissues might be minimized by using balanced crystalloid solutions for peritoneal washing.

## 1. Introduction

The omentum comprises two fatty structures. One apron-like structure drapes from the greater curvature of the stomach and extends to the transverse colon and splenic hilum, whereas the other thinner structure extends from the lesser curvature of the stomach to the liver [1,2]. Seventy-five percent of ovarian cancer patients present with advanced-stage metastatic disease that invariably involves the omentum (i.e., Stage III or Stage IV disease) [3]. Growth and coalescence of tumor implants on the omentum results in a bulky tumor mass (Figure 1). Bulky tumors on the omentum often angulate the bowel, causing substantial pain and leading to fatal bowel obstruction [3]. The Surveillance, Epidemiology, and End Results (SEER) Program of the National Cancer Institute estimates that 13,270 women will die from ovarian cancer in the United States in 2023 [4].

Omentectomy is the standard of care for metastatic ovarian cancer, but it is difficult to remove the omentum completely because of its anatomic location (Figure 1). Potential complications of omentectomy include injury to the transverse colon and spleen [5]. Furthermore, omentectomy may increase the risk of adhesive bowel obstruction and sepsis [6,7]. Omentectomy is also performed in patients with early-stage ovarian cancer as a preventative measure and for accurate staging, but this practice is not uniformly performed [5]. In a study of SEER data by McNally and colleagues, 4236 of 10,778 ovarian cancer patients (39.3%) with either Stage I disease (i.e., ovarian-confined disease) or Stage II disease (i.e., locally extended disease that is confined to the pelvis) had not undergone omentectomy [8].

There has been a lack of consensus about how much grossly normal omental tissue should be removed in ovarian cancer patients who present without overt metastasis, and the benefit of total omentectomy in these patients has been questioned by several investigators [5,8,9,10,11]. It has been suggested that random omental biopsies rather than total omentectomy may be sufficient for pathologic assessment where there is no overt metastasis [5,10,12]. Furthermore, two independent studies have found that omentectomy did not improve the survival of ovarian cancer patients who did not present with bulky omental disease [8,10]. In a study of rat xenograft models of ovarian cancer, Yokoyama and colleagues found that rats which underwent omentectomy concurrently with cancer cell inoculation had poorer survival rates than rats that underwent omentectomy after the omentum was involved with the tumor [9].

A caveat that needs to be considered is that disseminated ovarian cancer cells have a proclivity to implant on the omentum, and there are no effective strategies that minimize the risk of colonization of preserved omental tissues by occult cancer cells. Peritoneal washing using normal saline (0.9% sodium chloride) is commonly performed during ovarian cancer surgery, but the impact of this procedure on the disease has not been questioned. This review article focuses on the importance of reconsidering the use of normal saline for peritoneal washing in patients with ovarian cancer. Firstly, we discuss preclinical and clinical studies of the use of normal saline in fluid resuscitation (Section 2) and in peritoneal lavage (Section 3), highlighting that normal saline is not a biocompatible solution. We then discuss the biology of the omentum and omental metastasis (Section 4), and recent preclinical studies that reveal the mechanisms by which normal saline might render the omentum conducive for colonization by ovarian cancer cells (Section 5). Finally, we discuss the possibility that using balanced crystalloid solutions instead of normal saline for peritoneal washing might minimize the risk of colonization of preserved omental tissues (Section 6).

## 2. Normal Saline Is Not ‘Normal’: Insights from Fluid Resuscitation Studies

Normal saline is the most widely utilized crystalloid solution in medical practice and is commonly used for fluid resuscitation and lavage. It is thought that the medical use of saline solutions started during the cholera epidemic in the 1830s, but the composition of these early solutions did not resemble that of normal saline [13]. The earliest reference to normal saline appeared in an article by Lazarus-Barlow in 1896 that cited Hamburger for proposing 0.92% sodium chloride to be ‘normal’ for mammalian blood [14]. Compared to plasma and interstitial fluids, normal saline has a similar osmolality and a slightly higher (i.e., approximately 10% higher) sodium content [15,16,17,18,19,20,21] (Table 1). However, normal saline has a chloride content that is nearly 50% above physiological levels (Table 1). Furthermore, normal saline has a pH of 5.0 to 5.5, and does not contain a buffering agent (Table 1).

Fluid resuscitation with normal saline has been investigated in a number of clinical trials and observational studies in a variety of settings [19,20,21,22,23,24,25]. These studies have found that the main adverse effect of normal saline is hyperchloremic acidosis (Table 2). This acidosis is not caused by the low pH of normal saline per se but by the perturbation in acid-base equilibrium according to two models. Under the Henderson-Hasselbalch model, normal saline infusion leads to a reduction in pH by diluting the concentration of bicarbonate, the major buffering component in body fluids [26]. Using the Henderson–Hasselbalch model, Scheingraber and colleagues found that infusion with normal saline decreases the plasma bicarbonate concentration by 22% within two hours in patients undergoing gynecologic surgery [21]. Under the more recent Stewart model, infusion with normal saline leads to a reduction in pH by increasing the chloride ion concentration that decreases the plasma strong ion difference [19,20,21,22]. The strong ion difference of human plasma is approximately 40 mmol/L [21,22]. Using the Stewart model, Scheingraber and colleagues found that infusion with normal saline decreases the plasma strong ion difference to approximately 30 mmol/L within two hours [21]. 

Hyperchloremic acidosis in patients infused with normal saline is associated with an increased risk of renal and vascular dysfunction (Table 2). Preclinical and clinical studies have indicated that infusion with normal saline impairs renal function by decreasing renal blood flow and the glomerular filtration rate [22,27]. Specialized renal tubular epithelial cells called macula densa cells respond to increases in sodium or chloride ions by releasing adenosine triphosphate (ATP) [28]. In turn, ATP either activates ATP-binding purinergic receptors or degrades to form adenosine that activates the adenosine A1 receptor, leading to vasoconstriction of the afferent arteriole and reduction in the glomerular filtration rate [29]. 

Because of the adverse renal and vascular effects of hyperchloremic acidosis caused by normal saline, extensive clinical efforts have been directed to evaluating balanced crystalloids (i.e., balanced salt solutions) in fluid resuscitation. The two most-studied balanced crystalloids are lactated Ringer’s solution and Plasma-Lyte A. These solutions contain physiological concentrations of chloride and are buffered (Table 1). Lactated Ringer’s solution contains lactate whereas Plasma-Lyte A contains acetate, and both lactate and acetate are metabolized to form bicarbonate (Table 1). Scheingraber and colleagues found that, in contrast to normal saline, infusion with lactated Ringer’s solution did not decrease the plasma bicarbonate concentration or pH, and only modestly decreased the strong ion difference in patients undergoing gynecologic surgery [21]. A number of other clinical studies have reported that infusion using lactated Ringer’s solution or Plasma-Lyte A causes fewer adverse effects than normal saline and improves outcomes in both critically and non-critically ill adults [19,20,23,24,30,31,32]. 

**Table 2 ijms-24-16449-t002:** Deleterious effects of normal saline.

Type of Study	Reported Effects	References
Clinical (Intravenous infusion)	Electrolyte disturbance Hyperchloremic acidosis.	[21,22,23,24,25]
	Renal dysfunction Decrease in urine production; Decrease in renal blood flow; Decrease in renal cortical tissue perfusion; Renal failure.	[21,22,24,25,31,32]
	Cardiac dysfunction Myocardial ischemia; Myocardial infarction; Cerebral ischemia.	[25]
Pre-clinical (Intrarenal infusion)	Renal dysfunction Decrease in renal blood flow; Decrease in glomerular filtration rate.	[27]
Pre-clinical (Peritoneal lavage)	Decrease in pH	[33,34]
	Mesothelial cell damage Exfoliation of mesothelial cells; Formation of peritoneal adhesions.	[33,34,35]
In vitro (Primary culture of mesothelial cells)	Loss of cell adhesion	[33]
	Induction of oxidative stress	[36]
	Impairment of fibrinolytic properties	[34,36]

## 3. The Use of Normal Saline in Peritoneal Lavage and Its Impact on Mesothelial Cells

Normal saline is administered into the peritoneal cavity for diagnostic aspiration [37] and is also widely used to wash the cavity during abdominal surgeries [38]. The popularity of using normal saline for peritoneal lavage stems from a report by Burnett and colleagues in 1957 that this procedure was effective in treating peritonitis [39]. Peritoneal washes collected during ovarian cancer surgery are used for cytologic evaluation to assess the extent of the disease, and frequently contain sheets or aggregates of exfoliated mesothelial cells [40,41,42]. The mesothelium is composed of a monolayer of specialized cells that line and protect the peritoneal, pleural, and pericardial cavities and surfaces of internal organs [43]. Mesothelial cells are derived from the mesoderm and exhibit both epithelial-like and mesenchymal-like features [43]. Mesothelial cells in peritoneal washes can exhibit reactive changes such as enlarged hyperchromatic nuclei, prominent nucleoli, increased nuclear-to-cytoplasmic ratio, and dense cytoplasm, and it is difficult to distinguish reactive mesothelial cells from cancer cells based on cell morphology [40,41,42]. It has been reported that the rate of false-negative peritoneal washing cytology in ovarian cancer patients with disseminated peritoneal disease is 20% [40] and even higher [41]. 

It has been commonly thought that mesothelial cells are stripped by peritoneal washing due to shear stress from the procedure [41,42]. However, in a recent study [33], we found that exposure to normal saline downregulates the tight junction protein zonula occludens-1 (ZO-1) in mesothelial cells, causing these cells to detach (Table 2). A study of rats that underwent laparotomy found that peritoneal washing with normal saline promotes the formation of adhesions [35] (Table 2). Fibrin deposition is an early step in normal wound repair and is regulated by procoagulant and fibrinolytic molecules that are secreted by mesothelial cells such as tissue factor, the key initiator of coagulation [44], and tissue plasminogen activator (tPA) and urokinase plasminogen activator (uPA) that both cleave plasminogen to form plasmin which degrades fibrin clots [45]. Mesothelial cells also express plasminogen activator inhibitor-1 (PAI-1) that inhibits tPA and uPA [45]. Adhesion formation is initiated by an imbalance between fibrin deposition and fibrin degradation [43]. Połubinska and colleagues found that normal saline impairs the fibrinolytic properties of peritoneal mesothelial cells by generating reactive oxygen species (ROS) that increases the release of tissue factor and PAI-1 and suppresses the release of tPA [36]. These authors speculated that normal saline generates ROS in mesothelial cells by causing an influx of sodium that activates sodium/potassium-transporting ATPase [36]. However, acidosis caused by normal saline might also increase ROS. There is evidence that acidosis increases ROS production in other cell types [46]. 

Whereas previous studies of peritoneal lavage have largely focused on the effects of normal saline on the mesothelial lining, the impact of this procedure on underlying tissues is poorly understood. This merits investigation as more than one liter of solution is typically used for intraoperative washing [38] and approximately 25% of the instilled solution is not drained and dwells in the peritoneal cavity [34].

## 4. The Unique Biology of the Omentum and Omental Metastasis

The omentum is often termed an adipose tissue but has important protective functions, giving rise to its moniker the ‘abdominal policeman’ by Morison in 1906 [47]. The omentum contains an extensive vascular and lymphatic network [1,2], and has an immense absorptive ability [48]. Surgical studies in the 1960s exploited the absorptive ability of the omentum for therapeutic purposes by transposing this tissue to other sites to alleviate lymphedema [49]. Compared to other peritoneal fat depots such as mesenteric fat or gonadal fat, the omentum contains a higher abundance of fat-associated lymphoid clusters (FALCs) that were originally called ‘milky spots’ [50] (Figure 2). Omental FALCs are mainly composed of T cells, B cells and macrophages, and act as filters to clear pathogens from the peritoneal fluid [2,51]. Omental FALCs contain a glomerulus-like capillary network and high endothelial venules [2,51]. High endothelial venules in omental FALCs serve as a main entry pathway for neutrophil influx in response to peritoneal infection [52]. The importance of the omentum in clearing infections was recognized in a surgical discussion in 1902 that described the omentum as a ‘perfect sponge’ for septic material [53]. In a study of patients who underwent colorectal surgery with and without omentectomy, Ambroze and colleagues found that the incidence of post-operative sepsis was 2.5-fold higher in the omentectomy group [7].

Whereas metastasis of many other types of cancers occurs through hematogenous or lymphatic routes, metastasis of ovarian cancer is predominantly transcoelomic and occurs through the transport of cancer cells by interstitial fluid in the peritoneal cavity [3,54]. Cancer cells that shed into the peritoneal fluid are highly prone to lodge and form implants in omental FALCs [50,55]. Homing of ovarian cancer cells to the omentum is stimulated by C-C motif chemokine receptor 1 ligands and C-X-C motif chemokine receptor 1 ligands that are secreted by omental macrophages and adipocytes, respectively [56,57], and is also mediated by neutrophils. Ovarian cancer cells secrete interleukin-8, growth-regulated oncogene (GRO)-α, GRO-β, and granulocyte-colony stimulating factor that stimulate neutrophil chemotaxis and also induce neutrophils to expel chromatin fibers called neutrophil extracellular traps (NETs) [55]. It has been found that neutrophils mobilize into omental FALCs and form NETs prior to palpable metastasis in mice and women with early-stage ovarian cancer, and that NETs promote omental metastasis in part by physically trapping ovarian cancer cells [55]. 

Growth and invasiveness of ovarian cancer cells that implant on the omentum are stimulated by a repertoire of factors that mediate cross-talk between cancer cells and stromal cells. Omental adipocytes provide energy for tumor growth by transferring lipids to ovarian cancer cells through the lipid trafficking protein fatty acid-binding protein 4 [57]. Ovarian cancer cells secrete transforming growth factor (TGF)-β2 that acts in a paracrine manner on omental fibroblasts and adipose mesenchymal stem cells to induce expression of C-X-C motif chemokine ligand 12, interleukin-6, and vascular endothelial growth factor-A (VEGF-A) that in turn stimulate cancer cell growth and angiogenesis [58]. Ovarian cancer cells also secrete TGF-β1 that stimulates omental fibroblasts to express hepatocyte growth factor that promotes cancer cell invasion [59]. In addition, ovarian cancer cell-derived TGF-β1 stimulates mesothelial cells to produce fibronectin that promotes metastasis through interacting with integrin α5β1 on cancer cells [60]. Resident and recruited immune cells also foster a permissive environment for tumors on the omentum. A population of resident omental macrophages has been found to promote metastasis by inducing ovarian cancer cells to acquire a cancer stem cell-like phenotype [61]. Regulatory T cells are recruited to the omentum following the implantation of ovarian cancer cells and potently suppress anti-tumor immunity [62,63]. 

## 5. Impact of Normal Saline on the Omentum and Risk of Metastasis

Because of the absorptive ability of the omentum and its proclivity for transcoelomic metastasis, agents that are administered into the peritoneal cavity need to be evaluated for their impact on the omental microenvironment and for potentially stimulatory effects on metastasis. In a recent study using mouse models [33], we identified that the omentum has a prodigious ability to absorb normal saline administered into the peritoneal cavity and sustains significantly more mesothelial cell exfoliation than other peritoneal fat tissues. Notably, we found that the administration of normal saline stimulates the implantation of ovarian cancer cells onto the omentum through a series of remodeling events that is initiated by mesothelial cell exfoliation. A schematic representation of these events is shown in Figure 3. Specifically, normal saline induced expression of C-X3-C motif chemokine ligand 1 (CX3CL1) within and surrounding the vasculature in the omentum. CX3CL1 was found to be predominantly expressed as a membrane-bound protein in endothelial cells, and its induction caused monocytes/macrophages that express the cognate receptor C-X3-C motif chemokine receptor 1 (CX3CR1) to accumulate and persist until the omental mesothelium had repaired (Figure 3). Studies using genetically modified mice that express or lack functional CX3CR1 revealed that CX3CR1+ monocytes/macrophages promote the formation of tumor implants on the omentum following the administration of normal saline by stimulating neoangiogenesis [33] (Figure 3).

There are two ontogenically and phenotypically distinct types of macrophages that reside in the peritoneal cavity. It is thought that large peritoneal macrophages (LPM) arise from the yolk sac, whereas small peritoneal macrophages (SPM) derive from monocytes [64,65]. LPM lack major histocompatibility complex class II (MHCII) and express high levels of CD11b, F4/80 and intercellular adhesion molecule 2 (ICAM2), whereas SPM are MHCII+ and express lower levels of CD11b, F4/80 and ICAM2 than LPM [64,65]. CX3CR1+ monocytes/macrophages that accumulate in the omentum following the administration of normal saline exhibit features of SPM [33]. By contrast, it has been found that administration of normal saline does not increase LPM in the omentum and stimulates only a modest early influx of classical inflammatory monocytes that express Ly6C and C-C motif chemokine receptor 2 [33]. 

CX3CR1+ monocytes/macrophages have been found to mediate wound healing in various types of tissues [66,67]. Omental CX3CR1+ monocytes/macrophages constitutively express a wide repertoire of pro-angiogenic cytokines, and several cytokines such as C-C motif chemokine ligand 2 (CCL2), platelet-derived growth factor-B (PDGF-B), VEGF-A, and osteopontin (OPN) are further elevated following the administration of normal saline [33]. A common feature of these elevated cytokines is that their expression is positively regulated by nuclear factor kappa B (NF-κB) at the transcriptional level [68,69,70,71]. NF-κB activity depends on its translocation to the nucleus and nuclear localization of NF-κB increases in omental CX3CR1+ monocytes/macrophages following the administration of normal saline [33]. These findings suggest that the repair of the omental mesothelium following its exfoliation by normal saline is mediated by CX3CR1+ monocytes/macrophages that provide pro-angiogenic factors, of which several are upregulated through NF-κB activation induced by normal saline (Figure 3).

Normal saline decreases pH and low pH increases NF-κB activity in macrophages [72], but the precise mechanism is unclear. It has been reported that acidosis activates the NLR family pyrin domain containing 3 (NLRP3) inflammasome in macrophages, leading to secretion of interleukin-1β (IL-1β) [73]. IL-1β activates NF-κB [74]. It is therefore possible that acidosis caused by normal saline might increase NF-κB activity in macrophages through activating NLRP3-dependent IL-1β secretion. The mechanism by which normal saline induces CX3CL1 expression in omental endothelial cells is also unclear. As noted in Section 3, normal saline stimulates mesothelial cells to release tissue factor [36]. Tissue factor triggers a cascade that leads to the generation of thrombin, and thrombin induces CX3CL1 expression in endothelial cells [75]. It is therefore possible that the induction of CX3CL1 expression in endothelial cells could be intimately tied to the repair of the omental mesothelium following its exfoliation by normal saline (Figure 3). 

## 6. Could Balanced Crystalloids Be Used Instead of Normal Saline for Peritoneal Lavage?

In view of the findings that normal saline can render the omentum conducive for implantation of ovarian cancer cells, what other lavage solutions might be suitable for peritoneal washing? Several potential alternative solutions are listed in Table 3. Aqueous povidone–iodine (betadine) has been found to be more effective than normal saline in reducing the incidence of infectious complications when used for intraoperative peritoneal washing [76]. However, a study using a rat model of colorectal cancer has found that intraoperative washing with aqueous povidone–iodine compromises the integrity of the intestinal mucosa barrier by disrupting epithelial tight junctions which in turn causes endotoxin shock [77]. Furthermore, a recent study has identified that povidone–iodine causes toxicity to epithelial, endothelial, and mesothelial cells by releasing diatomic iodine that disrupts and depletes lipid rafts, causing loss of membrane integrity [78].

Conventional peritoneal dialysis solutions have been considered unsuitable for peritoneal lavage because these solutions contain glucose that generates glucose degradation products (GDPs) and advanced glycation end products (AGEs) [80]. Exposure to conventional peritoneal dialysis solutions disrupts membrane integrity of peritoneal mesothelial cells [79] (Table 3). It has also been reported that GDPs and AGEs can increase ROS production and impair tight junctions in peritoneal mesothelial cells [81,82]. One mechanism by which GDPs and AGEs might impair mesothelial cell tight junctions is through activating the p38 mitogen-activated protein kinase, p42/p44 mitogen-activated protein kinase, and protein kinase C pathways that induce expression of VEGF-A which in turn down-regulates ZO-1 [81,82]. Glucose-free peritoneal dialysis solutions are available but are more costly than conventional dialysis solutions [83]. 

Lactated Ringer’s solution and Plasma-Lyte A have been found in a number of independent clinical studies to cause fewer adverse effects than normal saline when used for fluid resuscitation [21,23,24,30,31,32] (Table 3). Lactated Ringer’s solution is comparable in cost to normal saline whereas the cost of Plasma-Lyte A is higher. In a recent study [33], we investigated the possibility of repurposing lactated Ringer’s solution for peritoneal lavage. In contrast to normal saline, administration of lactated Ringer’s solution into the peritoneal cavity of mice did not decrease the pH of interstitial fluid. It was found that lactated Ringer’s solution is less disruptive to mesothelial cell tight junctions than normal saline, and causes minimal exfoliation of mesothelial cells from the omentum. In contrast to normal saline, administration of lactated Ringer’s solution did not increase CX3CL1 expression in omental endothelial cells and did not cause accumulation of CX3CR1+ monocytes/macrophages in the omentum. Furthermore, lactated Ringer’s solution did not stimulate neoangiogenesis in the omentum. Notably, administration of lactated Ringer’s solution into the peritoneal cavity did not significantly increase lactate levels in the peritoneal fluid or peripheral blood, and did not stimulate the implantation of ovarian cancer cells onto the omentum or at other peritoneal sites [33].

Although no adverse effects were observed following the administration of lactated Ringer’s solution into the peritoneal cavity of mice [33], it remains to be determined whether Plasma-Lyte A might be a safer lavage solution than lactated Ringer’s solution. One difference between these two balanced crystalloids is their type of bicarbonate precursor. Lactated Ringer’s solution contains lactate whereas Plasma-Lyte A contains acetate (Table 1). Acetate metabolizes faster than lactate [84]. Plasma-Lyte A is considered more suitable for infusion where there is liver dysfunction because acetate is metabolized in many types of tissues whereas lactate is mainly metabolized in the liver [20,84]. Another advantage of Plasma-Lyte A is that it does not contain calcium (Table 1) which binds various drugs and is not compatible with anticoagulants [84]. However, the comparatively higher cost of Plasma-Lyte A needs to be considered in view of the volumes that are used for lavage.

## 7. Conclusions and Future Perspectives

For several decades, normal saline has been used for peritoneal washing in patients with ovarian cancer. However, this practice needs to be reconsidered. The presence of exfoliated mesothelial cells in peritoneal washings of patients with ovarian cancer and the difficulty in distinguishing reactive mesothelial cells from cancer cells have presented significant challenges for accurate cytologic evaluation. The use of balanced crystalloids instead of normal saline for peritoneal washing could potentially minimize the exfoliation of mesothelial cells and thereby improve the accuracy of cytologic assessment.

The use of normal saline for peritoneal washing in patients with ovarian cancer also needs to be carefully reconsidered in view of the potential impact of this procedure on the disease. Greater appreciation of the prodigious absorptive ability of the omentum is required. The administration of normal saline into the peritoneal cavity may elicit changes in the omentum that render its microenvironment conducive for colonization by occult ovarian cancer cells in the peritoneal fluid. Further investigation is needed to characterize changes in gene expression and signaling pathway activation in both immune and non-immune cell populations in the omentum following exposure to normal saline. Importantly, the possibility that the risk of omental colonization could be minimized by repurposing readily available and clinically used balanced crystalloids for peritoneal washing in patients with ovarian cancer merits further investigation.

## Figures and Tables

**Figure 1 ijms-24-16449-f001:**
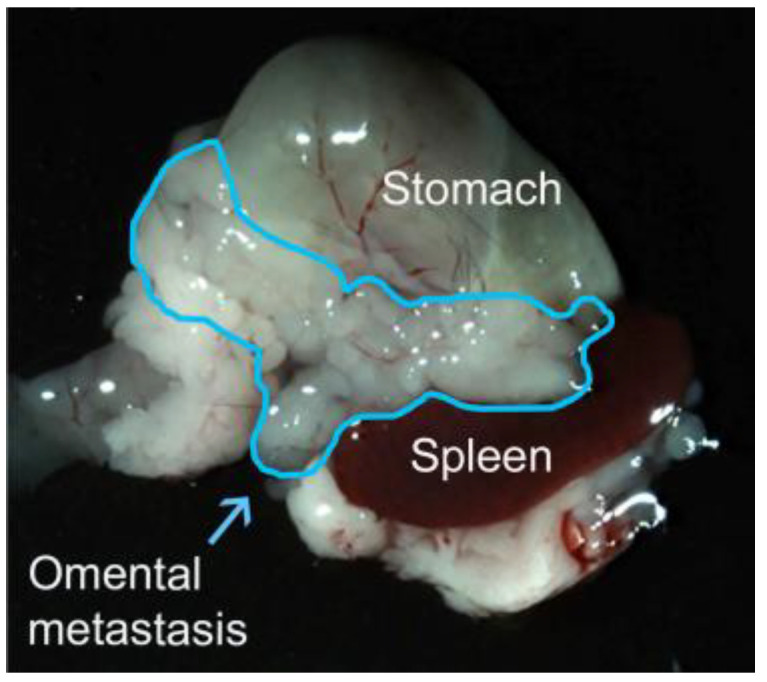
Omental metastasis in a syngeneic orthotopic mouse model of ovarian cancer. The bulky tumor mass involving the omentum is outlined in blue.

**Figure 2 ijms-24-16449-f002:**
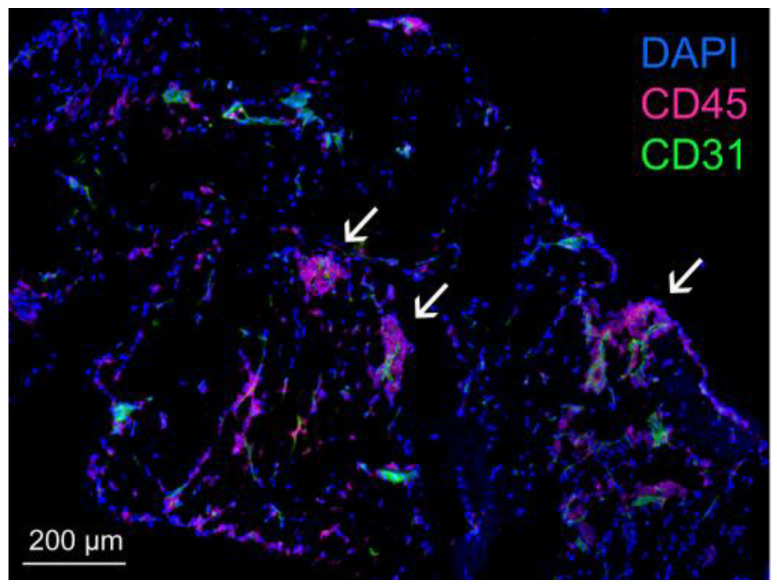
Immunofluorescence staining of normal omental tissue of an adult female C57BL/6 mouse, showing staining of the pan-leukocyte marker CD45 (red) and the endothelial cell marker CD31 (green). The tissue was counterstained with 4′,6-diamidino-2-phenylindole (blue). FALCs are indicated by arrows.

**Figure 3 ijms-24-16449-f003:**
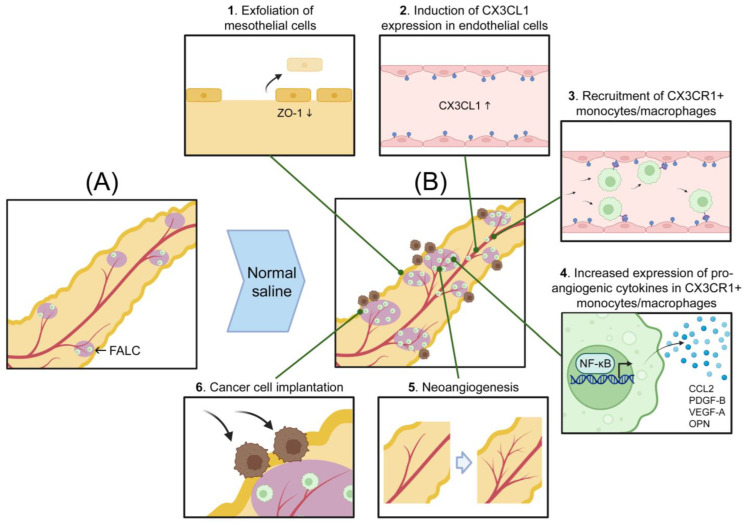
Normal saline remodels the omentum and stimulates its receptivity for ovarian cancer cells. Schematic representation of (**A**) the naïve omentum and (**B**) the omentum following administration of normal saline into the peritoneal cavity. Shown in (**B**) are the sequence of events following the absorption of normal saline by the omentum including (1) exfoliation of mesothelial cells through downregulation of the tight junction protein ZO-1, (2) induction of CX3CL1 expression in endothelial cells, (3) recruitment of CX3CR1+ monocytes/macrophages, (4) increased expression of pro-angiogenic cytokines in CX3CR1+ monocytes/macrophages through NF-κB activation, (5) stimulation of neoangiogenesis, and (6) implantation of ovarian cancer cells that circulate in the peritoneal fluid. This figure was created with BioRender.com (accessed on 6 October 2023).

**Table 1 ijms-24-16449-t001:** Composition of human body fluids, normal saline, and balanced crystalloids *.

	Plasma	Interstitial Fluids	Normal Saline	Lactated Ringer’s Solution	Plasma-Lyte A
Sodium (mmol/L)	136–145	136–145	154	130	140
Chloride (mmol/L)	102–110	108–118	154	109	98
Potassium (mmol/L)	3.5–5.7	3.5–5.0	0	4.0	5.0
Calcium (mmol/L)	1.0–2.5	1.2–2.8	0	2.7	0
Magnesium (mmol/L)	0.5–1.2	0.5–1.3	0	0	1.5
Bicarbonate (mmol/L)	24–25	22–28	0	0	0
Lactate (mmol/L)	<1.0	<1.2	0	28	0
Acetate (mmol/L)	Negligible	NA **	0	0	27
Gluconate (mmol/L)	Negligible	NA	0	0	23
pH	7.38–7.42	7.35–7.45	5.0–5.5	6.5–6.6	7.4
Osmolarity (mOsmol/L)	NA	NA	308	273	294
Osmolality (mOsmol/kg)	280–296	280–296	286	254	271

* Data compiled from references [15,16,17,18,19,20,21]. ** NA: not available/applicable.

**Table 3 ijms-24-16449-t003:** Alternatives to normal saline and their reported effects.

Type of Study	Type of Solution	Reported Effects	References
Clinical (Intravenous infusion)	Lactated Ringer’s solution	Lower incidence of hyperchloremic acidosis *	[21,23,32]
		Lower incidence of renal dysfunction *	[31,32]
		Lower C-reactive protein levels *	[30]
Clinical (Intravenous infusion)	Plasma-Lyte A	Lower incidence of hyperchloremic acidosis *	[24,32]
		Lower incidence of renal dysfunction *	[24,31,32]
Clinical (Peritoneal lavage)	Povidone-iodine	Lower incidence of infectious complications *	[76]
Pre-clinical (Peritoneal lavage)	Lactated Ringer’s solution	No decrease in pH	[33]
		Reduced degree of mesothelial cell exfoliation *	[33]
Pre-clinical (Peritoneal lavage)	Povidone-iodine	Formation of peritoneal adhesions	[35]
		Compromises integrity of intestinal mucosa barrier	[77]
In vitro	Lactated Ringer’s solution	Reduced loss of mesothelial cell adhesion *	[33]
	Povidone-iodine	Toxic to epithelial, endothelial and mesothelial cells	[78]
	Peritoneal dialysis solution	Loss of mesothelial cell membrane integrity	[79]

* As compared to normal saline.

## Data Availability

Not applicable.

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
