# Peer review of "Revisiting the Use of Normal Saline for Peritoneal Washing in Ovarian Cancer"

_ijms, 2023, doi:10.3390/ijms242216449_

Round 1

Reviewer 1 Report

Comments and Suggestions for Authors

In this work, the authors discussed the potential deleterious effects of the commonly used normal saline in peritoneal washing process during surgery. The authors drew insights from different aspects including the findings from fluid resuscitation studies and cytopathologic evaluation to point out the side effects of normal saline. In addition, the unique biology of omentum and disease mechanisms were highlighted. Finally they discussed the feasibility of using other balanced crystalloid solutions for peritoneal washing. Overall this manuscript could be potential interest of IJMS readers.

1. In section 4, the authors gave a thoroughly discussion on the unique biology of omentum and omental metastasis. It will be beneficial to shorten the description on biological mechanisms and highlight the discussion on interstitial fluid and its connection with normal saline.

2. The authors discussed several important findings from Ref [40] which support the side effects of normal saline during peritoneal washing process. I suggest the authors include one or two key figure/table into the review manuscript to consolidate the conclusions.

Reviewer 2 Report

Comments and Suggestions for Authors

The review “Revisiting the use of normal saline for peritoneal washing in ovarian cancer” by Akasaka, H. and Naora, H. is an interesting manuscript focusing on the significance of taking up for reconsideration the use of normal saline for peritoneal washing in patients with ovarian cancer.

The article has the following shortcomings:

1.      The title should be rewritten correctly in "TITLE" format, with capital letters where necessary.

2.      For the keywords: The authors did not adhere to MeSH for correctly choosing the keywords. There are keywords that are already in the title and therefore should NOT have been listed as keywords. Please, see lines 19-20.

3.      The abstract should be rewritten more rigorously and credible, so it should be reformulated in the light of the study applied, the papers included for revision, and the conclusions obtained.

4.      A better representation of the study and the work done is needed for the reader to demonstrate the results. A flow chart to describe the flow of information through the different phases of this review would be very welcome.

5.      A “List of Abbreviations” must be completed and reviewed carefully and may be better presented in a table format at the end of the article.

6.      A Graphical Abstract would be very welcome to increase the impact of this review and the IJMS's citations.

I congratulate the authors for their work.

Overall, the study is very interesting and deserves to be published with minor corrections. Overall, I recommend a minor revision.

I believe that after this minor revision provided by the authors on the issues suggested to be corrected and improved, it will provide useful and credible information for all readers, especially clinicians, and it is up to the Academic Editor to decide on its publication.

Thank you very much!

Comments on the Quality of English Language

A final reading on the MDPI platform by a native English speaker would be welcome.

Reviewer 3 Report

Comments and Suggestions for Authors
